# Association between prior tuberculosis disease and dysglycemia within an HIV-endemic, rural South African population

Alison C. Castle[1,2,3]*, Susanne S. Hoeppner[3], Itai M. Magodoro[4], Urisha Singh[1,5], Yumna Moosa[1,5], Ingrid V. Bassett[1,2,3], Emily B. Wong[1,5,6], Mark J. Siedner[1,2,3,5], on behalf of the Vukuzazi Study Team[¶]

1 Africa Health Research Institute, KwaZulu-Natal, South Africa, 2 Division of Infectious Diseases, Massachusetts General Hospital, Boston, Massachusetts, United States of America, 3 Harvard Medical School, Boston, Massachusetts, United States of America, 4 Emory Global Diabetes Research Center, Rollins School of Public Health, Emory University, Atlanta, Georgia, United States of America, 5 University of KwaZulu-Natal, Durban, KwaZulu-Natal, South Africa, 6 Division of Infectious Diseases, University of Alabama Birmingham, Birmingham, Alabama, United States of America

¶ Membership of the Vukuzazi Study Team is provided in the Acknowledgments.
* alison.castle@ahri.org

**Data Availability Statement:** Data and the data dictionary defining each field can be accessed at https://data.ahri.org/index.php/catalog/1006 via the Africa Health Research Institute Data Repository.

## Abstract

### Objective

Tuberculosis (TB) may predispose individuals to the development of diabetes. Such a relationship could have an outsized impact in high-prevalence TB settings. However, few studies have explored this relationship in populations heavily burdened by diabetes and TB.

### Methods

We analyzed data from a community-based population cohort that enrolled adults in rural South Africa. Individuals were considered to have prior TB if they self-reported a history of TB treatment. We fitted sex-specific logistic regression models, adjusted for potential clinical and demographic confounders, to estimate relationships between dysglycemia (HBA1c ≥6.5%) and prior TB. Propensity score-matched cohorts accounted for the differential age distributions between comparator groups. We examined the interactions between sex, prior TB, and HIV status.

### Results

In the analytic cohort (n = 17,593), the prevalence of prior TB was 13.8% among men and 10.7% among women. Dysglycemia was found in 9.1% of the population, and HIV prevalence was 34.0%. We found no difference in dysglycemia prevalence by prior TB (men OR 0.96, 95% CI 0.60–1.56: women OR 1.05, 95% CI 0.79–1.39). However, there was a qualitative interaction by HIV serostatus, such that among men without HIV, those with a history of TB had a greater prevalence of dysglycemia than those without prior TB (10.1% vs. 4.6%, p = 0.0077). An inverse relationship was observed among men living with HIV (prior TB 3.3% vs. no TB 7.3%, p = 0.0073).

Please email RDMServiceDesk@ahri.org. Access can be granted after publication and upon approval of the proposed analyses by the Vukuzazi Scientific Steering Committee and completion of a data access agreement.

**Funding:** Research reported in this publication was supported by the Fogarty International Center (D43 TW010543 [ACC]), National Institute of Mental Health, and the National Institute of Allergy and Infectious Diseases (T32 AI007433 [ACC], K24 AI141036 [IVB], K24 HL166024 [MJS]), of the National Institutes of Health. Additionally, this research was funded in part by Wellcome (Grant number Wellcome Strategic Core award: 201433/Z/16/A). For the purpose of open access, the author has applied a CC BY public copyright licence to any Author Accepted Manuscript version arising from this submission. The contents of this manuscript are solely the authors' responsibility and do not necessarily represent the official views of the funders. The funders of the study had no role in study design, data collection, data analysis, data interpretation, or writing of the report.

**Competing interests:** The authors have declared that no competing interests exist.

## Conclusions

Treated TB disease was not associated with dysglycemia in an HIV-endemic, rural South African population. However, we found a significant interaction between prior TB and HIV status among men, suggesting distinct pathophysiological mechanisms between the two infections that may impact glucose metabolism. Longitudinal studies are needed to better establish a causal effect and underlying mechanisms related to resolved TB, HIV, and diabetes.

## Introduction

Of the 10 million people who develop active tuberculosis (TB) annually, more than 80% survive after treatment. [1] In 2020, there were an estimated 155 million tuberculosis survivors prompting concerns about post-infectious sequelae and their impact on morbidity. [2, 3] While some of these sequelae have been identified as chronic lung disease, decreased exercise capacity, and hampered quality of life, there are other non-communicable disease consequences that may contribute to the observation that TB survivors have higher mortality compared to the general population. [4, 5] Specifically, cured TB may be associated with perturbations of glucose metabolism. In animal models, pulmonary TB results in the accumulation of serum free fatty acids and advanced glycation end products in tissue, which are central to the pathogenesis of diabetes. [6] Moreover, in a large cohort study from the United Kingdom, a prior TB diagnosis was associated with a two-fold higher incidence of diabetes. [7] Similar associations have been found among United States Veterans with latent TB disease diagnosed with tuberculin skin testing and interferon-γ release assays. [8]

Although this evidence suggests a relationship between tuberculosis and the future risk of diabetes, there remain critical gaps in our knowledge about this phenomenon. Most epidemiologic studies investigating TB and diabetes have occurred in low-TB prevalence areas outside of sub-Saharan Africa, where both the prevalence of TB and HIV might meaningfully impact the importance and mechanisms of this relationship. [7–13] Within South Africa, the province of KwaZulu-Natal is home to a significant burden of TB and HIV disease, with prevalence estimates of 26% and 34%, respectively. [14] Additionally, people living with HIV on antiretroviral therapy may have alterations in glucose metabolism from HIV infection, antiretroviral drugs, and/or weight gain from integrase inhibitor medications. [15] All these factors potentially complicate the relationship between treated TB and diabetes in HIV-endemic populations. Therefore, in South Africa and other countries with a high prevalence of these conditions, the relationship between post-TB sequelae and the development of dysglycemia is of high public health importance. [16]

In this study, we leveraged data from a cross-sectional, thoroughly characterized population cohort in rural KwaZulu-Natal to compare the prevalence of dysglycemia among individuals with and without prior TB. We then explored interactions between sex, previous TB, and HIV status. We hypothesized that dysglycemia would be higher among individuals with prior TB compared to those without TB.

## Methods

### Study population and procedures

We analyzed data from the Vukuzazi Study, a population-wide, cross-sectional study that enrolled residents in the uMkhanyakude District of KwaZulu-Natal, South Africa, as previously reported. [17] Briefly, all residents of the Africa Health Research Institute Demographic Health

Surveillance Site (AHRI DHSS) aged 15 years or older were visited at home and invited to participate in a mobile health screening that traveled through the study catchment area between May 2018—November 2019. The population in this region of South Africa is characterized as 100% of individuals of Black African descent, 58% of adults unemployed, and 66% with access to piped water in their homes. [14] Participants completed questionnaires on sociodemographic measures, current medications, and medical history, including previous TB treatment, diabetes, and HIV. All non-pregnant participants were screened for TB by digital chest x-ray. Sputum was collected from participants reporting TB symptoms or those with abnormal lung fields on chest x-ray as determined by real-time computer-assisted image analysis or by an experienced central radiologist. [18] Sputum samples were tested for *Mycobacterium tuberculosis* by Xpert® MTB/RIF Ultra test (Cepheid, Sunnyvale, USA) and liquid mycobacterial culture (BACTEC™ MGIT™ 960 System, Becton Dickinson, Berkshire, UK). Venipuncture whole blood samples were collected for hemoglobin A1c (VARIANT II TURBO Hemoglobin test system (Bio-Rad, Marnes-la-Coquette, France) and HIV (Genscreen Ultra HIV Ag-Ab enzyme immunoassay (Bio-Rad)). Participants with a positive HIV immunoassay had a reflex HIV-1 RNA viral load performed (Abbott RealTime HIV-1 Viral Load, Abbott, Illinois, USA).

## Study definitions

Our primary exposure of interest was prior tuberculosis based on self-reported history of at least one course of TB treatment. Participants with active TB, defined as a positive Xpert MTB/RIF, positive *Mycobacterium tuberculosis* culture, or individuals currently taking TB therapy at the time of study enrollment, were excluded from this analysis. We excluded those with active TB because active disease may result in stress hyperglycemia [19], and our work focuses on the effect of prior TB on the risk of future dysglycemia. We also excluded participants with active cancer or autoimmune diseases and those who reported taking anti-epileptic or glucocorticoid medications because these conditions and therapies can affect hemoglobin A1c. Time from TB diagnosis was calculated as the age of the last TB treatment course subtracted from the age at study enrollment in years. The number of TB episodes was defined as the number of completed 6-month treatment courses.

Our primary outcome of interest was dysglycemia, defined based on current WHO diagnostic thresholds as a hemoglobin A1c ≥6.5% (48.0mmol/mol). [20] Due to the cross-sectional nature of this study and because diabetes diagnoses require repeated testing over time, we used the term dysglycemia instead of diabetes. [21] Participants who self-reported a diagnosis of diabetes and reported use of diabetic medication within the last two weeks were also considered to have dysglycemia.

Participants with a positive HIV ELISA immunoassay were defined as having HIV. Waist circumference, measured in centimeters, was selected as the primary anthropometric measure used in our analyses because waist circumference has greater validity for identifying diabetes than other measures, such as body mass index. [22] Measurements for waist circumference were obtained directly on the skin or, when not possible, over light clothing, at the midpoint between the lower margin of the last palpable rib and the top of the iliac crest, at the end of normal expiration. Socioeconomic status was estimated using household asset ownership data collected within two years of the Vukuzazi enrollment date to generate a relative wealth index, as developed by Filmer and Pritchett. [23, 24]

## Statistical methods

Analyses were stratified by sex because of known differences in TB and diabetes prevalence among men and women. [14] We matched the characteristics of those with prior TB to

controls without prior TB to account for the differential age distribution between the two comparator groups. To do this, we first fitted logistic regression models among all AHRI DHSS adults with prior TB as the outcome of interest. Predictors for each sex-specific model included age, HIV status, smoking status, alcohol use, and socioeconomic status. The propensity score estimation for prior TB was then used to match controls with similar propensity scores (nearest neighbor matching, caliper 0.1). Next, we fitted multivariable logistic regression models of the propensity score-matched cohorts using frequency weights to evaluate the outcome of dysglycemia among those with and without prior TB in each sex-specific stratum. The following covariates were included: prior tuberculosis, age, waist circumference, HIV, smoking status, alcohol use, and socioeconomic status score. We evaluated two-way interactions between prior TB and HIV status for each model.

In sensitivity analyses, we fitted sex-specific multivariable logistic regression models for dysglycemia in the unmatched cohort. Additionally, we examined HIV-stratified subgroup analyses and fit multivariate linear regression models with hemoglobin A1c as a continuous outcome. Participants who reported the use of diabetic medication within the last two weeks (n = 167) were excluded from the linear regression analyses because hemoglobin A1c values would be modified in participants taking medication. Lastly, we assessed whether time from TB diagnosis and number of TB treatment courses were associated with dysglycemia among those with prior TB. All sensitivity analyses were adjusted for potential clinical and demographic confounders listed above. Statistical analyses were conducted in Stata (Version 17, StataCorp, College Station, Texas, USA).

### Inclusivity in global research

Additional information regarding the ethical, cultural, and scientific considerations specific to inclusivity in global research is included in the Support Information (S1 Checklist).

### Ethical considerations

The institutional review boards approved the Vukuzazi study at the University of KwaZulu-Natal Biomedical Research Ethics Committee (BREC) and Mass General Brigham (Protocol 2018P001802). All participants gave written informed consent to participate.

### Results

Of the eligible adults within the catchment area (n = 34,721), 18,041 (51.9%) were enrolled in the Vukuzazi study. A total of 17,952 (99.5%) completed hemoglobin A1c testing. Of these participants, we excluded 245 (1.4%) with active tuberculosis, 105 (0.6%) taking antiepileptics, 7 (<0.1%) with active cancer, 1 (<0.1%) with autoimmune disease, and 1 (<0.1%) taking prednisone resulting in the final analytic sample of 17,593 (S1 Fig). Of the eligible participants within AHRI's DHSS, women (OR 2.03 [95% CI 1.94–2.12]; p<0.001) and individuals over the age of 50 (OR 2.01 [95% CI 1.92–2.11]; p<0.001) were more likely to attend the Vukuzazi Study compared to men and younger participants.

The study population included 58.7% women (n = 11,986) with an overall mean age of 40.5 years (SD 19.5) (Table 1). The prevalence of dysglycemia with a hemoglobin A1c of ≥6.5% or active use of diabetes therapy was 5.0% among men (n = 282) and 11.0% among women (n = 1,316). HIV prevalence was 34.0% (n = 6,096), with 82.4% of people living with HIV having a suppressed viral load on antiretroviral therapy. Prior tuberculosis was more prevalent in men (13.8%) than women (10.7%) (Table 1). Compared to people without a history of TB, participants with a history of TB were more likely to be older (men 47.0 vs. 34.3 years; women

**Table 1. Unweighted baseline characteristics of participants.**

| Characteristic | Total Cohort n = 17,593 | Men n = 5,607 | Women n = 11,986 |
|---|---|---|---|
| Sex (%) | N/A | 41.3 | 58.7 |
| Age (year) | 40.5 ± 19.5 | 36.2 ± 18.9 | 42.5 ± 19.4 |
| Mean BMI (kg/m2) | 27.6 ± 7.4 | 23.5 ± 4.8 | 29.6 ± 7.6 |
| Mean Waist Circumference(cm) | 87.6 ± 16.6 | 80.0 ± 12.4 | 91.1 ± 17.1 |
| Dysglycemia- no.(%) | 1,598 (9.1) | 282 (5.0) | 1,316 (11.0) |
| Controlled Diabetes- no.(%)* | 120 (0.7) | 29 (0.5) | 91 (0.8) |
| HIV positive- no.(%) | 6,096 (34.0) | 1,406 (24.4) | 4,690 (38.5) |
| Controlled HIV- no.(%)^ | 5,019 (82.4) | 1,031 (73.3) | 3,988 (85.1) |
| History of TB treatment- no.(%) | 1,996 (11.7) | 760 (13.8) | 1,236 (10.7) |
| Time from TB diagnosis (years) | 11.1 ± 9.3 | 10.3 ± 9.3 | 11.7 ± 9.3 |
| Number of TB treatment courses | 1.4 ± 2.0 | 1.5 ± 2.5 | 1.3 ± 1.5 |
| Active TB- no.(%) | 245 (2.6) | 111 (3.2) | 134 (2.3) |
| Socioeconomic status# | 0.23 ± 2.0 | 0.25 ± 2.0 | 0.22 ± 2.0 |
| Past or Active Smoker- no.(%) | 1,447 (8.1) | 1,275 (22.1) | 172 (1.4) |
| Consumes Alcohol- no.(%) | 2,262 (12.6) | 1,609 (27.8) | 653 (5.4) |

* Persons taking diabetic medication with controlled HBA1c

^ Controlled disease defined as HIV positive on treatment and viral load <400 copies/uL

# Socioeconomic score ranges from -7.0 to +7.0

45.6 vs. 42.2 years), living with HIV (men 60.3% vs. 18.5%; women 75.7% vs. 34.0%), and with lower socioeconomic scores (men -0.04 vs. 0.30; women 0.04 vs. 0.24) (S1 Table). The propensity score-matched cohorts achieved balance on key characteristics based on the standardized differences of <0.25, variance ratios between 0.5–2.0, and kernel density plots (S2 Fig, S2 and S3 Tables). [25, 26]

In the propensity score-matched univariate analyses, there were no differences in dysglycemia prevalence by prior TB history among men (no TB 6.6% vs prior TB 5.3%, OR 0.79, 95% CI 0.51–1.23) or women (no TB 10.8% vs prior TB 9.6%, OR 0.88, 95% CI 0.67–1.14). When adjusting for age, waist circumference, HIV, smoking status, alcohol use, and socioeconomic score, again, prior TB status was not associated with dysglycemia (men prior TB OR 0.96, 95% CI 0.59–1.56: women prior TB OR 1.05, 95% CI 0.79–1.39). However, we found a significant qualitative interaction between prior TB disease and HIV serostatus (Table 2, Fig 1). Among men without HIV, individuals with prior TB had a dysglycemia prevalence of 10.1%, significantly higher than men without prior TB (4.6%, p-value 0.0077). In contrast, dysglycemia among men living with HIV was more prevalent in individuals without a prior TB (7.3% vs. 3.3%, p-value 0.0073). The qualitative effect of HIV status, prior TB, and dysglycemia was not present among women (Fig 1).

We found similar relationships between prior TB and dysglycemia in sensitivity analyses using both the unmatched and HIV-stratified cohorts. Among individuals with prior TB, 43.2% (n = 864) reported the year of TB diagnosis with an average time of 11.1 years (SD 9.3) before study enrolment. In univariate analyses, time since TB diagnosis was associated with dysglycemia (OR 1.04, 95% CI 1.02–1.07), but not after adjusting for clinical and demographic confounders (OR 1.01, 95% CI 0.98–1.04). The total number of TB treatment courses per individual was 1.4 (SD 2.0), and it was not associated with dysglycemia prevalence in univariate (p = 0.185) or multivariate regression analyses (p = 0.302).

**Table 2. Propensity score matched multivariate logistic regression models for dysglycemia.**

| Characteristic | Odds Ratio | p-value | 95% Confidence Interval | |
|---|---|---|---|---|
| Male Model | | | | |
| Age (years) | 1.039 | <0.001 | 1.021 | 1.057 |
| Waist Circumference (cm) | 1.059 | <0.001 | 1.040 | 1.079 |
| HIV status | 3.115 | 0.002 | 1.500 | 6.466 |
| Prior TB | 2.698 | 0.008 | 1.295 | 5.620 |
| Past or Active Smoker | 0.590 | 0.151 | 0.287 | 1.212 |
| Never Alcohol | 0.358 | 0.003 | 0.184 | 0.698 |
| Socioeconomic score | 1.075 | 0.250 | 0.950 | 1.216 |
| Prior TB*HIV | 0.147 | <0.001 | 0.053 | 0.402 |
| Female Model | | | | |
| Age (years) | 1.046 | <0.001 | 1.035 | 1.058 |
| Waist Circumference (cm) | 1.036 | <0.001 | 1.027 | 1.046 |
| HIV status | 0.891 | 0.607 | 0.573 | 1.385 |
| Prior TB | 1.157 | 0.562 | 0.707 | 1.892 |
| Past or Active Smoker | 0.782 | 0.695 | 0.229 | 2.668 |
| Never Alcohol | 0.475 | 0.067 | 0.214 | 1.052 |
| Socioeconomic score | 1.035 | 0.369 | 0.960 | 1.115 |
| Prior TB*HIV | 0.860 | 0.623 | 0.470 | 1.571 |

Prior TB*HIV represents the interaction term for these two variables

## Discussion

In a large, well-characterized population in rural South Africa, we did not find a relationship between a history of treated TB and dysglycemia in the general population. Specifically, the prevalence of dysglycemia measured at the time of study enrollment was similar between participants who did and did not report past TB treatment. This null result is informative because it contrasts with a growing body of experimental work conducted in the United Kingdom and United States populations that suggests that a prior TB diagnosis may predispose individuals to develop diabetes. In South Africa, where there is comparatively a greater burden of tuberculosis, our results suggest the potential metabolic consequences of TB disease may not be clinically nor statistically significant at a population level.

Our study differs from published data related to TB and diabetes likely because previous work in this area used different study designs and evaluated different populations. For example, other studies typically measured glucose metabolism at the time of active disease and shortly after completion of TB treatment. [10, 11, 27, 28] Few studies have examined long-term diabetes incidence after TB disease. [7] Pearson et al., in one of the largest studies to date (n = 8065), albeit retrospective, found a nearly two-fold increase in the incidence of diabetes among individuals with prior TB compared to those without TB over a 6-year observation period in the United Kingdom. [7] Our cross-sectional study measured hemoglobin A1c at enrollment, which for nearly half of individuals with prior TB, was 11 years after their TB diagnosis and treatment course. Additionally, in contrast to studies conducted in the Western populations, our cohort was set in a low-resourced rural area with a high burden of HIV. Reflecting the population of rural South Africa, the mean age of our cohort was relatively young (men 36.2 years; women 42.5 years) compared to high-income settings. [29] This also likely resulted in a comparatively lower prevalence of dysglycemia. Type 2 diabetes is determined by a multitude of factors such as ethnicity, socioeconomic status, food security, nutrition, quality of available foods, cultural behaviors, and physical inactivity. [30–32] Many of

## Dysglycemia Prevalence in Men

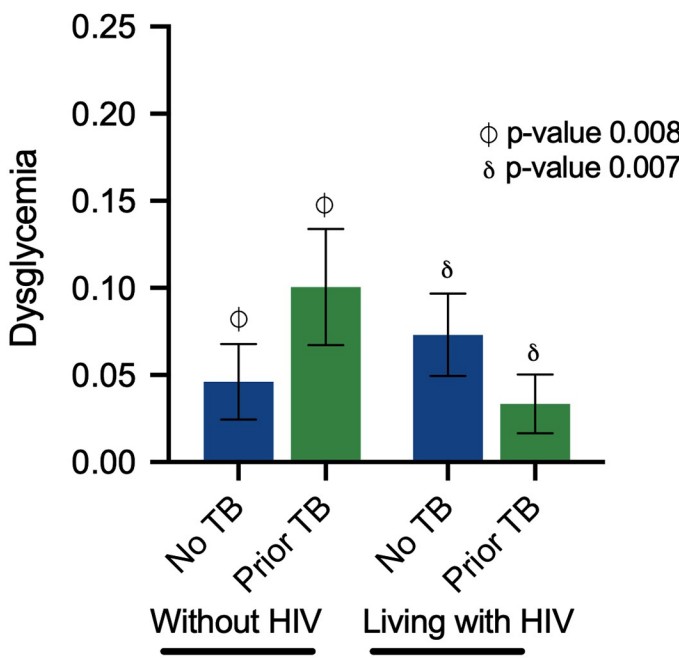

## Dysglycemia Prevalence in Women

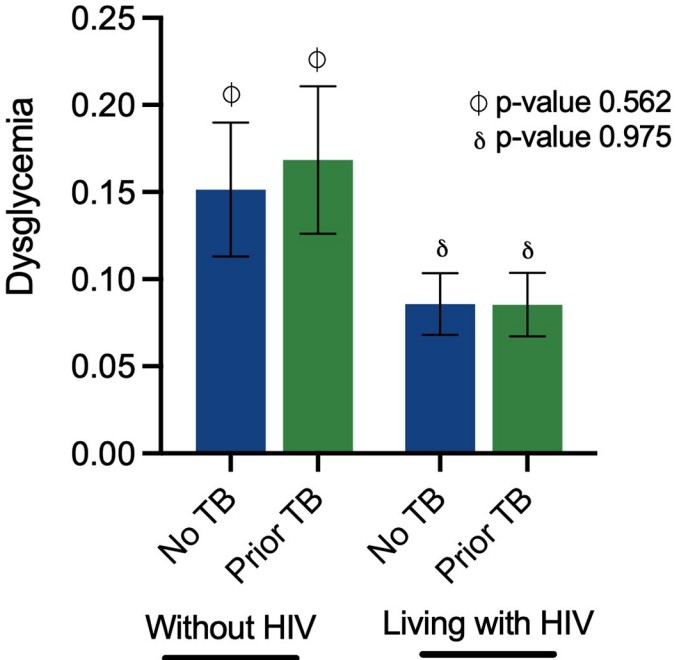

**Fig 1. Prior TB and HIV interactions stratified by sex.** Among men without HIV (top), dysglycemia prevalence is higher in those with prior history of TB disease compared to men without prior TB. The inverse relationship was found in men living with HIV, where dysglycemia prevalence was lower in those with prior TB. Women (bottom) did not have an association between HIV status, prior TB, and dysglycemia prevalence.

these factors differ between our study population and the populations where longitudinal tuberculosis and diabetes studies have been conducted, which may explain the differences in our results. Within South Africa, Tamuhla et al. used data from a longitudinal health system to assess the relationship between prior TB disease on hemoglobin A1c values among persons living with diabetes. [33] This retrospective study found no evidence that TB disease influences the trajectory of glycemic control among persons with known diabetes over five years. [33] While our study population and objectives differed, our results that prior TB disease is not associated with dysglycemia at a population level are consistent with the results of this South African study. [33]

Our results suggest that HIV may modify the relationship between prior TB and diabetes. Among men without HIV, dysglycemia was more prevalent in individuals with a history of TB. This relationship was inversed among individuals living with HIV, such that men living with HIV and prior TB had a lower prevalence of dysglycemia compared to those with no history of TB. It is plausible that this qualitative interaction contributed to our null findings in the general population. Namely, the increased dysglycemia prevalence among men without HIV and prior TB may have been muted by the substantially lower dysglycemia prevalence among men living with HIV and prior TB. This implies that future studies of TB and diabetes should consider sex- and HIV-specific stratification. Notably, HIV-negative persons with active TB have different pro-inflammatory cytokine responses compared to persons living with HIV and TB. [34, 35] We hypothesize that these differential inflammatory responses and their potential metabolic effects may contribute to the HIV-subgroup differences in our study.

Unexpectedly, we found that, among men living with HIV, individuals with a history of prior TB had lower dysglycemia prevalence compared to those without TB. This may be due to residual active paucibacillary or extrapulmonary TB that may have been undetected on sputum screening and chest x-rays. It may also be indicative of other infectious or non-infectious etiologies leading to malnutrition, weight loss, and poor overall health status. Additionally, there are conflicting data about the accuracy of hemoglobin A1c testing among specific subgroups that could affect the results of our study. Although not consistent across all African populations [36], hemoglobin A1c testing has been found to be falsely low in individuals living with HIV and persons with acute TB infection in certain studies. [37–40] The potential error in detecting diabetes using this clinical assay could explain our results. Further work is needed to validate hemoglobin A1c testing in individuals living with HIV and prior TB.

In addition to active TB, untreated latent TB disease (LTBI) has also been associated with a future risk of diabetes. A retrospective cohort study among 574,113 U.S. Veterans found a greater incidence of diabetes among those with LTBI. [8] Tuberculin skin testing and interferon-γ release assays were unavailable in our study. Still, previous South African studies estimate the prevalence of LTBI ranging from 34–89%, with exceptionally high rates among people living with HIV, miners, and healthcare workers. [41, 42] These findings suggest that our control comparator group likely included a non-ignorable proportion of individuals with latent TB disease. We may not have found a significant difference in our results because the controls, with high LTBI prevalence, may have a predisposition to aberrant glucose metabolism at baseline. Elucidating the clinical spectrum of TB disease and diabetes incidence is needed to understand the at-risk subgroups for more targeted diabetes screening.

The strengths of our study include a large population-based cohort that was thoroughly characterized with testing for HIV, hemoglobin A1c, chest x-rays, sputum cultures, GeneXpert, and TB treatment data to identify and differentiate individuals with prior TB. Furthermore, anthropometric measures, socioeconomic data, and individual behaviors were accounted for in our analyses, which are variables that often confound the bidirectional relationship between tuberculosis and diabetes. There were also important limitations. The cross-

sectional study design prevents the ability to make causal inferences between prior TB and dysglycemia. Individuals with prior TB have higher mortality compared to the general population; therefore, the people with prior TB who survived to participate in our study had greater overall health compared to the total population of people with prior TB, which could result in sampling bias. In our raw, unmatched sample, individuals with prior TB were, on average older than those without a history of TB. We attempted to mitigate this limitation by using propensity score matching and, in doing so, achieved balance in terms of measured confounders between groups. We also might have misestimated our exposure. Despite the well-characterized population cohort, the diagnosis of prior TB is inherently challenging to establish. We may have misidentified individuals with prior TB based on self-report because TB treatment can be initiated empirically without microbiological confirmation in low-resource settings, [43] and because participants may incorrectly remember or misreport their medical and treatment history, particularly for a condition like TB, which is stigmatized in South Africa. [44]

In conclusion, we did not find a simple association between self-reported prior tuberculosis and dysglycemia as measured by hemoglobin A1c in a large, HIV-endemic rural South African population. However, we found an interactive effect of HIV status and prior TB that indicated that men without HIV and a prior TB diagnosis had higher dysglycemia compared to their counterparts without TB, but the opposite relationship is present among men living with HIV. Future prospective longitudinal observational studies in sub-Saharan populations that carefully define states of TB, HIV, and diabetes across the lifespan will be required to determine the relationships and mechanisms underpinning this complex relationship.

## Supporting information

**S1 Fig. Analytic cohort flow diagram.**
(PDF)

**S2 Fig. Sex-stratified original cohorts versus matched cohorts based on age.**
(PDF)

**S1 Table. Baseline cohort characteristics by sex and prior TB status.**
(PDF)

**S2 Table. Standardized differences between raw and matched cohort.**
(PDF)

**S3 Table. Weight applied to each matched control.**
(PDF)

**S1 Checklist. Inclusivity in global research.**
(PDF)

## Acknowledgments

We thank the residents of the Africa Health Research Institute demographic surveillance area and members of the Vukuzazi Study Team who collected these data.

Emily Wong is the lead author for the Vukuzazi Study Team and may be contacted at Emily.wong@ahri.org. The members of the Vukuzazi Study Team are: Mark Siedner (AHRI, principal investigator), Thumbi N'dungu (AHRI, principal investigator), Willem Hanekom (AHRI, principal investigator), Kobus Herbst (AHRI, principal investigator), Dickman Gareta (AHRI, Head of Research Data Management), Tshwaraganang Modise (AHRI, Data management), Jaco Dreyer (AHRI, Data management), Siyabonga Nxumalo (AHRI, Data

management), Sweetness Dube (AHRI, Data documentation), Resign Gunda (AHRI, programme management), Ashmika Surujdeen (AHRI, programme management), Olivier Koole (AHRI, programme management), Ngcebo Mhlongo (AHRI, study clinician), Sanah Bucico (AHRI, lead nurse), Thandeka Khoza (AHRI, Clinical research), Theresa Smit (AHRI, Laboratory), Greg Ording (AHRI, Laboratory), Innocentia Mpofana (AHRI, Diagnostic Laboratory Manager), Khadija Khan (AHRI, Biorepository Manager), Zizile Sikhosana (AHRI, Somkhele Laboratory Supervisor), Sashen Moodley (AHRI Microbiology Laboratory Supervisor), Hollis Shen (AHRI, Head: Exploratory Research Division), Philippa Mathews (AHRI, Clinical Governance), Nompilo Buthelezi (AHRI, Training Coordinator), Hlolisile Khumalo (AHRI, Nursing Manager), Sanah Bucibo (AHRI, Professional Nurse), Nozipho Mbonambi (AHRI, Professional Nurse), Hloniphile Ngubane (AHRI, Professional Nurse),Thokozani Simelane (AHRI, Professional Nurse), Khanyisani Buthelezi (AHRI, Professional Nurse), Sphiwe Ntuli (AHRI, Professional Nurse), Nombuyiselo Zondi (AHRI, Professional Nurse), Siboniso Nene (AHRI, Professional Nurse), Bongumenzi Ndlovu (AHRI, Enrolled Nurse), Talente Ntimbane (AHRI, Enrolled Nurse), Mbali Mbuyisa (AHRI, Enrolled Nurse), Xolani Mkhize (AHRI, Enrolled Nurse), Melusi Sibiya (AHRI, Enrolled Nurse), Ntombiyenkosi Ntombela (AHRI, Enrolled Nurse), Mandisi Dlamini (AHRI, Enrolled Nurse), Hlobisile Chonco (AHRI, Enrolled Nurse), Hlengiwe Dlamini (AHRI, Enrolled Nurse), Doctar Mlambo (AHRI, Enrolled Nurse), Nonhlanhla Mzimela (AHRI, Enrolled Nurse), Zinhle Buthelezi (AHRI, Enrolled Nurse), Zinhle Mthembu (AHRI, Enrolled Nurse), Thokozani Bhengu (AHRI, Enrolled Nurse), Sandile Mthembu (AHRI, Enrolled Nurse), Phumelele Mthethwa (AHRI, Enrolled Nurse),Zamashandu Mbatha (AHRI, Enrolled Nurse), Welcome Petros Mthembu (AHRI, Enrolled Nurse), Anele Mkhwanazi (AHRI, Clinical Research Assistant Supervisor), Mandlakayise Zikhali (AHRI, Clinical Research Assistant Supervisor), Phakamani Mkhwanazi (AHRI, Clinical Research Assistant), Ntombiyenhlanhla Mkhwanazi (AHRI, Clinical Research Assistant), Rose Myeni (AHRI, Clinical Research Assistant), Fezeka Mfeka (AHRI, Clinical Research Assistant), Hlobisile Gumede (AHRI, Clinical Research Assistant), Nonceba Mfeka (AHRI, Clinical Research Assistant), Ayanda Zungu (AHRI, Clinical Research Assistant), Nonhlanhla Mfekayi (AHRI, Clinical Research Assistant), Smangaliso Zulu (AHRI, Clinical Research Assistant), Mzamo Buthelezi (AHRI, Clinical Research Assistant), Senzeni Mkhwanazi (AHRI, Clinical Research Assistant), Mlungisi Dube (AHRI, Clinical Research Assistant), Hosea Kambonde (iMarketing Consultants, IT Systems Developer), Lindani Mthembu (AHRI, Information Technology Assistant), Seneme Mchunu (AHRI, Information Technology Assistant), Sibahle Gumbi (AHRI, Research Admin Assistant), Tumi Madolo (AHRI, Research Data Manager), Thengokwakhe Nkosi (AHRI, Driver), Sibusiso Mkhwanazi (AHRI, Driver), Sibusiso Nsibande (AHRI, Driver), Mpumelelo Steto (AHRI, Driver), Sibusiso Mhlongo (AHRI, Driver), Velile Vellem (Aurum Innova (Pty) Ltd, Driver), Pfarelo Tshivase (Aurum Innova (Pty) Ltd, Driver), Jabu Kwinda (Aurum Innova (Pty) Ltd, Driver), Bongani Magwaza (AHRI, General Worker), Siyabonga Nsibande (AHRI, General Worker), Skhumbuzo Mthombeni (AHRI, General Worker), Sphiwe Clement Mthembu (AHRI, General Worker), Antony Rapulana (AHRI, Laboratory Technologist), Jade Cousins (AHRI, Laboratory Technologist), Thabile Zondi (AHRI, Laboratory Technologist), Nagavelli Padayachi (AHRI, Laboratory Technologist), Freddy Mabetlela (AHRI, Laboratory Technologist), Simphiwe Ntshangase(AHRI, Laboratory Technician/LIMS Administrator), Nomfundo Luthuli (AHRI, Laboratory Technician), Sithembile Ngcobo (AHRI, Laboratory Technologist), Kayleen Brien (AHRI, Laboratory Technologist), Sizwe Ndlela (AHRI, Laboratory Technician), Nomfundo Ngema (AHRI, Laboratory Technician), Nokukhanya Ntshakala (AHRI, Laboratory Technician), Anupa Singh (AHRI, Laboratory Technician), Rochelle Singh (AHRI, Laboratory Technician), Logan Pillay (AHRI, Laboratory Technician), Kandaseelan Chetty (AHRI, Laboratory Technician),

Ashentha Govender (AHRI, Laboratory Technician), Pamela Ramkalawon (AHRI, Laboratory Research Technician), Nondumiso Mabaso (AHRI, Laboratory Intern), Kimeshree Perumal (AHRI, Laboratory Intern), Senamile Makhari (AHRI, Biorepository Laboratory Technician), Nondumiso Khuluse (AHRI, Biorepository Laboratory Technician), Nondumiso Zitha (AHRI, Biorepository Research Assistant), Hlengiwe Khathi (AHRI, Biorepository Research Assistant), Mbuti Mofokeng (AHRI, Clinical Specimen Driver/Laboratory Assistant), Nomathamsanqa Majozi (AHRI, Public Engagement), Nceba Gqaleni(AHRI, Public Engagement),Hannah Keal (AHRI, Communications), Phumla Ngcobo (AHRI, Communications), Costa Criticos (AHRI, Operational Oversight), Raynold Zondo (AHRI, Operational Oversight), Dilip Kalyan (AHRI, Operational Oversight), Clive Mavimbela (AHRI, Operational Oversight), Anand Ramnanan (AHRI, Procurement), Sashin Harilall (AHRI, Grants Office), Kennedy Nyamande (University of KwaZulu-Natal, Pulmonology Consultant), Jaikrishna Kalideen (Perumal and Partners Radiologist Inc, Radiologist), Ramesh Jackpersad (Jacpersand Inc, Radiologist), Kgaugelo Moropane (Aurum Innova (Pty) Ltd, Radiographer), Boitsholo Mfolo (Aurum Innova (Pty) Ltd, Radiographer), Khabonina Malomane (Aurum Innova (Pty) Ltd Radiographer).

## Author Contributions

**Conceptualization:** Alison C. Castle, Emily B. Wong, Mark J. Siedner.

**Data curation:** Alison C. Castle.

**Formal analysis:** Alison C. Castle, Susanne S. Hoeppner, Emily B. Wong, Mark J. Siedner.

**Funding acquisition:** Alison C. Castle.

**Investigation:** Mark J. Siedner.

**Methodology:** Alison C. Castle, Susanne S. Hoeppner, Emily B. Wong, Mark J. Siedner.

**Supervision:** Mark J. Siedner.

**Validation:** Susanne S. Hoeppner.

**Writing – original draft:** Alison C. Castle.

**Writing – review & editing:** Susanne S. Hoeppner, Itai M. Magodoro, Urisha Singh, Yumna Moosa, Ingrid V. Bassett, Emily B. Wong, Mark J. Siedner.

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
