## [Decision Letter · Decision Letter 0]

1 Nov 2022

PONE-D-22-26235Association between prior tuberculosis disease and dysglycemia within an HIV-endemic, rural South African populationPLOS ONE

Dear Dr. Castle,

Thank you for submitting your manuscript to PLOS ONE. After careful consideration, we feel that it has merit but does not fully meet PLOS ONE’s publication criteria as it currently stands. Therefore, we invite you to submit a revised version of the manuscript that addresses the points raised during the review process.

Both reviewers concurred that this was generally a well-written paper which added to the knowledge of the subject. There were a few minor comments: 1. Not all eligible residents participated - this should be explained 2. The age of the participants was low and they would be expected to have a low prevalence of Diabetes and this should be discussed as a limitation There is also a recommendation of certain references to consult

We look forward to receiving your revised manuscript.

Kind regards,

Elizabeth S. Mayne, M.D.

Academic Editor

PLOS ONE

Journal Requirements:

4. One of the noted authors is a group or consortium the Vukuzazi Study Team. In addition to naming the author group, please list the individual authors and affiliations within this group in the acknowledgments section of your manuscript. Please also indicate clearly a lead author for this group along with a contact email address.

Additional Editor Comments:

Both reviewers concurred that this was generally a well-written paper which added to the knowledge of the subject. There were a few minor comments:

1. Not all eligible residents participated - this should be explained

2. The age of the participants was low and they would be expected to have a low prevalence of Diabetes and this should be discussed as a limitation

There is also a recommendation of certain references to consult

Reviewers' comments:

Reviewer's Responses to Questions

**Comments to the Author**

1. Is the manuscript technically sound, and do the data support the conclusions?

Reviewer #1: Yes

Reviewer #2: Yes

2. Has the statistical analysis been performed appropriately and rigorously? 

Reviewer #1: I Don't Know

Reviewer #2: Yes

3. Have the authors made all data underlying the findings in their manuscript fully available?

Reviewer #1: Yes

Reviewer #2: Yes

4. Is the manuscript presented in an intelligible fashion and written in standard English?

Reviewer #1: Yes

Reviewer #2: Yes

5. Review Comments to the Author

Reviewer #1: The authors report the relationship between prior tuberculosis infection and dysglycaemia in a population in rural South Africa. The paper is generally well written with sound methodology and contributes to the literature on tuberculosis and dysglycaemia.

Minor comments

1. Approximately half of the eligible residents in the DHSS participated in the study. It would be useful to know the basic characteristics of those who did not take part, so the reader can make an assessment of the generalisability of the results

2. The mean age of the study population is rather low and the background prevalence of diabetes would also be expected to be fairly low. the authors should comment on this limitation.

3. Reference 1 is only cited after several other references-this should be corrected.

4. The word "comprise" is not followed by "of"

5. The references should be formatted in a standardised fashion

Reviewer #2: There are a number of African studies that have shown that the association of TB and diabetes varies with the diagnostic test used.

This may have relevance for your findings

1. Eur Respir J. 2017 Jul; 50(1): 1700004.

2.. J Infect Dis 2016; 213: 1163–1172

3.Int J Tuberc Lung Dis 2017 Feb 1;21(2):208-213

6. PLOS authors have the option to publish the peer review history of their article (what does this mean?). If published, this will include your full peer review and any attached files.

Reviewer #1: No

Reviewer #2: **Yes: **Jaya A George

---

## [Author Response · Author response to Decision Letter 0]

2 Dec 2022

Dr. Elizabeth S. Mayne

Academic Editor 

PLOS ONE

2nd December 2022

Dear Dr. Mayne and editorial team, 

Thank you for inviting us to submit a revised draft of our manuscript entitled "Association between prior tuberculosis disease and dysglycemia within an HIV-endemic, rural South African population" to PLOS ONE. We appreciate the time and effort you and each reviewer have dedicated to providing insightful feedback to strengthen our paper. We followed the reviewers’ suggestions and have outlined the revisions to the paper below. We hope that our edits and the responses we provide below satisfactorily address all the issues and concerns you and the reviewers have noted. 

JOURNAL REQUIREMENTS:

RESPONSE: The title page, abstract, and manuscript have been edited with tracked changes to meet all of the style requirements instructed above. 

2. Please include a complete copy of PLOS’ questionnaire on inclusivity in global research in your revised manuscript. 

RESPONSE: A section on Inclusivity in global research is now included in our methods section, and a complete questionnaire is attached as a supplement (S4 Checklist). 

3. In your data availability statement, you have not specified where the minimal data set underlying the results described in your manuscript can be found. PLOS defines a study's minimal data set as the underlying data used to reach the conclusions drawn in the manuscript and any additional data required to replicate the reported study findings in their entirety. All PLOS journals require that the minimal data set be made fully available. Upon re-submitting your revised manuscript, please upload your study’s minimal underlying data set as either Supporting Information files or to a stable, public repository and include the relevant URLs, DOIs, or accession numbers within your revised cover letter. For a list of acceptable repositories, please see http://journals.plos.org/plosone/s/data-availability#loc-recommended-repositories. Any potentially identifying patient information must be fully anonymized.

RESPONSE: Our current data statement is updated to include the following bolded words:

“Data and the data dictionary defining each field can be accessed at https://data.ahri.org/index.php/catalog/1006 via the Africa Health Research Institute Data Repository. Please email RDMServiceDesk@ahri.org. Access can be granted after publication and upon approval of the proposed analyses by the Vukuzazi Scientific Steering Committee and completion of a data access agreement.” 

4. One of the noted authors is a group or consortium the Vukuzazi Study Team. In addition to naming the author group, please list the individual authors and affiliations within this group in the acknowledgments section of your manuscript. Please also indicate clearly a lead author for this group along with a contact email address.

RESPONSE: The individual authors and their affiliations within the Vukuzazi Study Team are now listed within the Acknowledgments section. Emily Wong is the lead author for the Vukuzazi Study Team and may be contacted at Emily.wong@ahri.org.

5. Please review your reference list to ensure that it is complete and correct. If you have cited papers that have been retracted, please include the rationale for doing so in the manuscript text or remove these references and replace them with relevant current references. Any changes to the reference list should be mentioned in the rebuttal letter that accompanies your revised manuscript. If you need to cite a retracted article, indicate the article’s retracted status in the References list and also include a citation and full reference for the retraction notice.

RESPONSE: All references were reviewed, and no papers have been retracted. 

REVIEWER #1 COMMENTS:

Thank you for your time and effort in giving us feedback. We address your comments one by one in the following. 

1. Approximately half of the eligible residents in the DHSS participated in the study. It would be useful to know the basic characteristics of those who did not take part, so the reader can make an assessment of the generalisability of the results

RESPONSE: Lines 198-200 now include the basic demographic information (sex and age) of the eligible participants who did not enroll in the Vukuzazi Study. We state, “Of the eligible participants within AHRI’s DHSS, women (OR 2.03 [95% CI 1.94-2.12]; p<0.001) and individuals over the age of 50 (OR 2.01 [95% CI 1.92-2.11]; p<0.001) were more likely to attend the Vukuzazi Study compared to men and younger participants.”

2. The mean age of the study population is rather low and the background prevalence of diabetes would also be expected to be fairly low. The authors should comment on this limitation.

RESPONSE: Whereas the study population is representative of the rural region where our study was conducted, we acknowledge that the age distribution of this profile is younger than in high-income countries and therefore resulted in a relatively lower prevalence of diabetes. The following statement is now added to lines 271-273. “Reflecting the population of rural South Africa, the mean age of our cohort was relatively young (men 36.2 years; women 42.5 years) compared to high-income settings [44]. This also likely resulted in a comparatively lower prevalence of dysglycemia.”

3. Reference 1 is only cited after several other references-this should be corrected.

RESPONSE: This was an error with EndNote that has now been resolved. 

4. The word "comprise" is not followed by "of"

RESPONSE: Line 202. The statement “was comprised of” was changed to “included”. 

5. The references should be formatted in a standardised fashion.

RESPONSE: References are organized using EndNote using the PLoS style format. 

REVIEWER 2 COMMENTS:

1. There are a number of African studies that have shown that the association of TB and diabetes varies with the diagnostic test used. This may have relevance for your finding 1. Eur Respir J. 2017 Jul; 50(1): 1700004. 2.. J Infect Dis 2016; 213: 1163–1172. 3.Int J Tuberc Lung Dis 2017 Feb 1;21(2):208-213.

RESPONSE: We appreciate the reviewer bringing these studies to our attention. The diagnostic tests used in all three of these studies were used to detect acute TB infection. By contrast, considered the effect of previously treated TB and dysglycemia. Nonetheless, these studies raise important questions about the validity of hemoglobin A1c testing for diagnosing diabetes in key populations and we have incorporated two of them into lines 305-307. 

Thank you for giving us the opportunity to strengthen our manuscript with your valuable comments and queries. We’ve incorporated your feedback and hope the revised manuscript will be suitable for publication in PLOS ONE. 

Sincerely,

Alison Castle 

Corresponding Author

Africa Health Research Institute

719 Umbilo Rd, Durban, Kwa-Zulu Natal, 4001

Alison.castle@ahri.org

---

## [Decision Letter · Decision Letter 1]

19 Jan 2023

PONE-D-22-26235R1Association between prior tuberculosis disease and dysglycemia within an HIV-endemic, rural South African populationPLOS ONE

Dear Dr. Castle,

Thank you for submitting your manuscript to PLOS ONE. After careful consideration, we feel that it has merit but does not fully meet PLOS ONE’s publication criteria as it currently stands. Therefore, we invite you to submit a revised version of the manuscript that addresses the points raised during the review process.

We look forward to receiving your revised manuscript.

Kind regards,

Elizabeth S. Mayne, M.D.

Academic Editor

PLOS ONE

Journal Requirements:

Additional Editor Comments:

Please just correct the minor editing errors in the references.

Reviewers' comments:

Reviewer's Responses to Questions

**Comments to the Author**

1. If the authors have adequately addressed your comments raised in a previous round of review and you feel that this manuscript is now acceptable for publication, you may indicate that here to bypass the “Comments to the Author” section, enter your conflict of interest statement in the “Confidential to Editor” section, and submit your "Accept" recommendation.

Reviewer #1: (No Response)

Reviewer #2: All comments have been addressed

2. Is the manuscript technically sound, and do the data support the conclusions?

Reviewer #1: Yes

Reviewer #2: Yes

3. Has the statistical analysis been performed appropriately and rigorously? 

Reviewer #1: Yes

Reviewer #2: Yes

4. Have the authors made all data underlying the findings in their manuscript fully available?

Reviewer #1: Yes

Reviewer #2: Yes

5. Is the manuscript presented in an intelligible fashion and written in standard English?

Reviewer #1: Yes

Reviewer #2: Yes

6. Review Comments to the Author

Reviewer #1: My only additional comment regards the formatting of references. The authors should be consistent in the use of upper case letters in the titles of their references. In some instances sentence case is used while in others, every word in the title begins with an upper case letter.

Reviewer #2: (No Response)

7. PLOS authors have the option to publish the peer review history of their article (what does this mean?). If published, this will include your full peer review and any attached files.

Reviewer #1: **Yes: **Alisha N Wade MBBS DPhil

Reviewer #2: **Yes: **Professor J George

---

## [Author Response · Author response to Decision Letter 1]

20 Jan 2023

Dr. Elizabeth S. Mayne

Academic Editor 

PLOS ONE

20th January 2023

Dear Dr. Mayne and editorial team, 

Thank you for inviting us to submit a revised draft of our manuscript entitled "Association between prior tuberculosis disease and dysglycemia within an HIV-endemic, rural South African population" to PLOS ONE. We followed the reviewer's suggestions and have outlined the revisions to the paper below. We hope that our edits and the responses we provide below satisfactorily address all the issues and concerns you and the reviewers have noted. 

REVIEWER #1 COMMENTS:

My only additional comment regards the formatting of references. The authors should be consistent in the use of upper case letters in the titles of their references. In some instances sentence case is used while in others, every word in the title begins with an upper case letter.

RESPONSE: 

We appreciate you highlighting the discrepancies in our reference titles. The following references have now been reformatted so that all are in sentence case only: 5, 8, 11, 12, 15, 16, 17, 20, 21, 22, 23, 25, 28, 30, 31, 32, 33, 34. 

Thank you for giving us the opportunity to strengthen our manuscript with your valuable comments. We’ve incorporated your feedback in the revised manuscript and hope this will be suitable for publication in PLOS ONE. 

Sincerely,

Alison Castle 

Corresponding Author

Africa Health Research Institute

719 Umbilo Rd, Durban, Kwa-Zulu Natal, 4001

Alison.castle@ahri.org

---

## [Decision Letter · Decision Letter 2]

14 Feb 2023

Association between prior tuberculosis disease and dysglycemia within an HIV-endemic, rural South African population

PONE-D-22-26235R2

Dear Dr. Castle,

We’re pleased to inform you that your manuscript has been judged scientifically suitable for publication and will be formally accepted for publication once it meets all outstanding technical requirements.

Kind regards,

Elizabeth S. Mayne, M.D.

Academic Editor

PLOS ONE

Additional Editor Comments (optional):

Reviewers' comments:

Reviewer's Responses to Questions

**Comments to the Author**

1. If the authors have adequately addressed your comments raised in a previous round of review and you feel that this manuscript is now acceptable for publication, you may indicate that here to bypass the “Comments to the Author” section, enter your conflict of interest statement in the “Confidential to Editor” section, and submit your "Accept" recommendation.

Reviewer #1: All comments have been addressed

2. Is the manuscript technically sound, and do the data support the conclusions?

Reviewer #1: (No Response)

3. Has the statistical analysis been performed appropriately and rigorously? 

Reviewer #1: (No Response)

4. Have the authors made all data underlying the findings in their manuscript fully available?

Reviewer #1: (No Response)

5. Is the manuscript presented in an intelligible fashion and written in standard English?

Reviewer #1: (No Response)

6. Review Comments to the Author

Reviewer #1: (No Response)

7. PLOS authors have the option to publish the peer review history of their article (what does this mean?). If published, this will include your full peer review and any attached files.

Reviewer #1: No

---

## [Editor Report · Acceptance letter]

6 Mar 2023

PONE-D-22-26235R2 

Association between prior tuberculosis disease and dysglycemia within an HIV-endemic, rural South African population 

Dear Dr. Castle:

I'm pleased to inform you that your manuscript has been deemed suitable for publication in PLOS ONE. Congratulations! Your manuscript is now with our production department. 

Kind regards, 

on behalf of

Dr. Elizabeth S. Mayne 

Academic Editor

PLOS ONE